# On-line Writing of Fiber Bragg Grating Array on a Two-mode Optical Fiber for Sensing Applications

**DOI:** 10.3390/ma12081263

**Published:** 2019-04-17

**Authors:** Haihu Yu, Wenjing Gao, Xin Jiang, Huiyong Guo, Shan Jiang, Yu Zheng

**Affiliations:** 1National Engineering Laboratory for Fiber Optic Sensing Technology, Wuhan University of Technology, Wuhan 430070, China; hhyu@whut.edu.cn (H.Y.); gaowenjing@whut.edu.cn (W.G.); jiangx@whut.edu.cn (X.J.); ghylucky@163.com (H.G.); 2Wuhan WUTOS Co., Ltd., Wuhan 430070, China; jiangshan@wutos.com

**Keywords:** two-mode fiber, fiber Bragg grating, on-line grating writing, temperature sensing, curvature sensing

## Abstract

On-line fabricated fiber Bragg grating (FBG) array and its sensing potentials have attracted plenty of attention in recent years. In this paper, FBG arrays are written on-line on a two-mode fiber, and this two-mode fiber Bragg grating (TM-FBG) is further experimentally investigated for temperature and curvature sensing. The responses of this sensor were characterized by 11.2 pm/°C and −0.21 dB/m^−1^ for temperature and curvature, respectively. Based on the measurements, a dual-parameter fiber sensing system was developed, which can realize the quasi-distributed, simultaneous detection of temperature and curvature, making it suitable for structural health monitoring or perimeter security.

## 1. Introduction

Fiber Bragg grating (FBG), as an important fiber technology, possesses advantageous properties for fiber-based sensors such as high resistance to electromagnetic interference, enhanced sensitivity, compact size and desirable network extensions [1,2]. Various physical quantities such as temperature, stress, curvature and refractive index have been well studied, using FBG-based fiber sensing devices [3,4,5]. A typical FBG is applied on single-mode fibers (SMFs), inscribed with periodic Bragg grating structure, so as called a single-mode FBG. Single-mode FBG is widely used in the fiber sensing field, which has characteristics of low transmission loss and single mode propagation [6]. Not only limited to SMFs, grating structures, in principle, can be written on any fiber with a sensitive core, for instance, a few-mode fiber. A few-mode fiber supports a limited number of transverse modes [7]. It has been demonstrated that pressure, stress and curvature sensing can be realized individually using a few-mode fiber based sensing system [8,9,10,11]. If now we consider combining it with the FBG technology, a few-mode FBG, therefore, it can be expected to be used for multi-parameter measurements and comparatively reduced cross-sensitivity, as observed in single-mode FBG [12,13,14,15]. 

Recently, FBG arrays and distributed sensing networks are attracting broad interests in the field of optical fiber sensing, from both industry and academic. Long-distance, large-capacity FBG array on SMFs have been demonstrated, using a FBG on-line written system [16,17]. In principle, the same technology can be applied to few-mode fibers. Unfortunately, due to technical difficulties, it has not been demonstrated. On the other hand, due to low transmission loss, few-mode FBG can be considered for long-distance signal transmission and multi-point distributed measurements. 

In this paper, using the on-line FBG writing technique, two-mode fiber Bragg gratings (TM-FBGs) and a TM-FBG array are successfully fabricated. We then use the TM-FBG for sensing studies. Temperature and curvature sensing are experimentally investigated. Furthermore, based on the TM-FBG array, a dual-parameter fiber sensing system is developed, which can realize the quasi-distributed simultaneous detection of temperature and curvature. To make the system practical, the potential application of TM-FBG array is discussed.

## 2. The Principle of the Two-Mode FBG

Firstly, we checked the transmission properties of a two-mode fiber in theory. Transverse modes, supported by an optical fiber, are determined by the structural parameters of the fiber, as well as the operating wavelength. In a step-index fiber, the normalized cut-off frequency *V_c_* is given by:(1)$$V_{c}=\frac{2\pi }{\lambda_{0}}a\sqrt{n_{1}^{2}-n_{2}^{2}},$$
where *λ*_0_ is wavelength, *a* is core radius of the fiber, and *n*_1_ and *n*_2_ are the refractive indices of the core and cladding glasses. Several lowest order linearly polarized (LP) modes and their corresponding *V_c_* are given in Table 1 [18].

From Table 1, when the *V_c_* is within 2.405 and 3.823, only the two LP_01_ and LP_11_ modes are supported. Based on Equation (1), the *V_c_*, the core radius (*a*) and the difference of refractive index between the core and cladding materials (*n*_1_ and *n*_2_) can be adjusted. We then used the COSMOL Multiphysics to make the finite element method (FEM) calculations, in order to figure out the expected fiber parameters. 

We first checked the standard SMF G.652. The refractive indices of the fiber core *n*_1_ and cladding *n*_2_ are set as 1.450 and 1.445; the initial fiber diameter *2r*_2_ is 125 μm, and the core diameter *2r*_1_ is 8.3 μm. The working wavelength is 1550 nm. The results are shown in Figure 1. When the fiber diameter increases from 125 μm to 240 μm, the core diameter ramps up from 8.3 μm to 15.9 μm gradually. The fiber starts to support the LP_01_ and LP_11_ modes, when its outer diameter is larger than 175 μm. The co-existence of the two modes is kept, when the fiber diameter is between 175 and 240 μm. The LP modes are superpositions of vector modes namely HE and TE/TM modes, for example, the LP_11_ mode, is four-fold degeneracy of the HE_21_, TE_01_ and TM_01_ modes, as clearly depicted in the results of the numerical simulations, as seen in Figure 1.

Furthermore, we analyzed the characteristics of FBG in the two-mode fiber. Based on the coupled-mode theory of FBG, modes coupling happens between the forward and backward propagating core modes in FBG, when the absolute values of the propagation constants are identical. The reflection spectra of a single-mode FBG with a fiber outer diameter of 125 μm and a two-mode FBG with a fiber outer diameter of 200 μm were investigated. The grating period Λ was set at 535.59 nm, and the reflection spectra were calculated by transfer matrix method based on coupled-mode theory. The results are plotted in Figure 2. Figure 2a and c show the curve of the mode propagation constant and reflection spectrum of the single-mode FBG. According to the calculations (Figure 2a), the propagation constant of the LP_01_ mode decreases monotonically with the increase of wavelength, while the grating resonance is independent of the wavelength change. The propagation constant and the grating resonance have only one intersection (marked as A) at the wavelength of 1550.90 nm, corresponding to the central wavelength (marked as A’) of FBG in the refection spectrum, as shown in Figure 2c. Figure 2b shows the relationship of calculated propagation constants and resonance condition of the LP_01_ mode, LP_11_ mode and the coupled mode in two-mode fiber, of which the grating resonance wavelength are marked as B, C and D, corresponding to the three wavelength of TM-FBG in Figure 2d. Due to the co-existence of the LP_01_ and LP_11_ modes in the fiber, each mode has a specific wavelength satisfying the grating resonance condition, which are at wavelengths 1549.90 nm (B’) and 1552.00 nm (D’), respectively. In addition, mode coupling exists between the two LP modes, which forms a coupled-mode [19]. This coupled-mode in TM-FBG is generated by the coupling between the forward mode of LP_01_ mode and the backward mode of LP_11_ mode or the forward mode of LP_11_ mode and the backward mode of LP_01_ mode. The propagation constant of the coupled mode is approximately one half of the sum, of those the two LP modes. The resonance wavelength of the coupled mode is at around 1550.95 nm (C’). Based on the above analysis, it is found that this TM-FBG has three characteristic reflections, and they are considered for later sensing experiments.

The first order high order mode (HOM) LP_11_ is prone to fiber bending, as compared to the fundamental mode (FM) LP_01_. Using COSMOL, the confinement loss is calculated by [20]:(2)αloss=8.686×2πλ[μm]×Im(neff)×109[dB/km]
where the Im(*n_eff_*) is the imaginary part of the effective refractive index, of a supported mode. The confinement losses of the LP_01_ and LP_11_ modes, under the bending curvatures ranging from 35 m^−1^ to 100 m^−1^, were calculated (Figure 3). It can be seen that the confinement loss of the LP_01_ mode (black dots) remained small (~10^−4^ dB/m), while the loss of the LP_11_ mode (red dots) increased significantly from 0.25 to 0.6 dB/m, when the curvature ramped up from 35 m^−1^ to 100 m^−1^. The LP_11_ mode is deemed as a leaky mode when fiber is bent. Due to this feature, by carefully identifying the reflection peak of the LP_11_ mode, the TM-FBG can be possibly used for curvature sensing. In addition, by measuring the shift of the reflection wavelength of the LP11 mode, temperature or strain sensing can also be realized simultaneously, using the same TM-FBG.

## 3. Online Writing of FBG Array

FBG arrays inscribed with various gratings can be written by using the on-line writing system on our customized drawing tower. Figure 4a shows the schematic diagram of the drawing tower, affiliated with the FBG on-line writing system. A fiber preform with a germanium-doped, photosensitive core is fed continuously into a graphite resistance furnace. After heated to over 1900 °C, the preform is to be drawn into bare fibers. With precise control of the parameters such as furnace temperature, drawing and feeding speed, fiber tension and preform position, the final fiber can be produced with expected dimensions. Before the fiber goes through a double-coating and UV curing unit, there is a FBG writing platform which FBG array can be written on the fiber. After being written, the fiber is then directed to the collection wheel on the ground floor. Using the on-line writing technique, a large number of FBGs can be continuously inscribed in a single fiber piece of a long distance (e.g., 50 km), without complex post-processing steps such as coating decortication, grating inscription and fiber-splicing, as required by conventional FBG writing. This technique brings advantages such as highly consistent grating reflectivity, increased mechanical strength and reduced splicing loss. 

FBG arrays can be written in real time during fiber drawing. Figure 4b shows the FBG writing stage. An excimer laser operating at 193 nm wavelength acts as the UV irradiation source, of which the laser spot size is 6 mm × 12 mm. The pulse width is ~10 ns, and the maximum repetition rate is 500 Hz. The maximum available pulse energy is ~10 mJ. A phase mask is placed ~0.1 mm next to the bare fiber, with a mask size of 10 mm × 10 mm. The laser beam goes through three cylindrical lenses and is then focused onto the phase mask with beam line of 0.7 mm × 10 mm. Through the phase mask, ±1^st^ interference fringes are generated and expose to the bare fiber to inscribe FBG. The period of grating Λ is determined by:(3)$$\Lambda =\frac{\Lambda_{\mathrm{mask}}}{2},$$
where Λ_mask_ is the period of the phase mask.

Based on this system, a two-mode FBG array was successfully written on a step-index fiber with a photosensitive core. The optical fiber preform for FBG inscription was the glass preform, with a low amount of germanium, less than 0.6%, doped in the fiber core and the numerical aperture (NA) of 0.14. The fiber possessed the same structure as the standard single-mode optical fiber G.652. The pitch of the phase mask was 1071.93 nm and the laser pulse energy was 8 mJ. The preform used for fiber drawing was a standard one for SMF with a fiber core diameter 8.3 um, and we used the same preform for the fabrication of the two-mode (TM) fiber. Based on our theoretical calculations, it was found for a fiber to support two modes, the core diameter needs to be expanded to 13.3 um, so as the fiber outer diameter to be ~200 um. By controlling the preform feeding and fiber drawing rates, the diameter of fiber was increased from 125 μm to 200 μm, for the reason to draw a TM fiber. To avoid the occurrence of excessive shaking and breaking in grating fabrication process, the drawing speed was controlled at 10–15 m/min and the average drawing tension at 10–30 N. Experimental reflection spectra of the FBG with the fiber diameter of 125 μm and 200 μm are shown respectively in Figure 5a,b, which are in good agreement with the theoretical plots in Figure 2c,d. 

## 4. Temperature and Curvature Sensing

The TM-FBG based fiber sensor was characterized for temperature and curvature sensing. The temperature experimental setup is shown in Figure 6. As seen from the figure, we selectively chose a fiber section with FBG inscription as the temperature unit. Figure 7a exhibits the red-shift of the center wavelength of three reflections in the TM-FBG, when the temperature increased from 30 °C to 100 °C with a step of 10 °C. The central wavelengths of the reflections of the LP_01_ and LP_11_ modes under different temperature are plotted in Figure 7b. The results showed that both the LP_01_ and the LP_11_ modes had a similar temperature response of 11.2 pm/°C, owing to the same transmission medium (fiber core), which had the same thermo-optical and expansion coefficients of the two modes. Indeed, the temperature response of the two-mode fiber was very close to the value of temperature response in a single-mode FBG (10.7 pm/°C) [21]. 

The calculation in Figure 3 indicates that the mode confinement loss of the mode LP_01_ was much lower than that of the LP_11_ mode, when the fiber was bent. We then considered using the TM-FBG for bending measurements. An amplified spontaneous emission source (ASE, ASE-CL-20-B, WavePhotonics, Pittsfield, USA), an optical spectrum analyzer (OSA, AQ6370B, Yokogawa, Tokyo, Japan) and a circulator were used for the measurements. The experimental setup is depicted in Figure 8. The optical fiber was wrapped around a loop with an adjustable radius. The smaller the loop radius, the severer the bending applied to the TM fiber. As seen from the figure, we selectively chose a fiber section without FBG inscription as the bending unit, and this section was bent into a circle of different diameters. Figure 9a shows the reflection spectrum when the bending diameter changed from 67 mm to 26 mm. Figure 9b presents the reflected power as a function of the bending curvature, of the two modes. Only when the curvature was larger than 25 m^−1^, the change of the transmission became obvious. From Figure 9b, when the curvature increased from 25 m^−1^ to 80 m^−1^, the intensity of the reflection peak of the LP_11_ mode decreased at a rate of −0.21 dB/m^−1^, and the linearity was 0.991. On the contrary, the reflected peak intensity of the LP_01_ mode was almost kept unchanged at −60.932 dBm. Considering the actual intensity of the light source will fluctuate in the process, the variation of the reflection peak intensity of mode LP_01_ as the reference, it can modify the reflection peak results of mode LP_11_, making the measurement of the fiber curvature more accurate. Figure 9c shows the difference change of reflectivity intensity between the two peaks with the increase of curvature. The near-field mode patterns were recorded using an infrared camera (Bobcat-320, Xenics) and plotted in Figure 9d. Clearly, we can observe that the LP_11_ mode was susceptible to the fiber bending, and therefore, led to a dimmed output.

## 5. TM-FBG Array and Its Application

Using the on-line FBG writing system on the drawing tower, a 100-meter-long two-mode FBG array with an equal grating spacing of 1 meter was inscribed. The total of the 100 FBGs were continuously written in a row without any splicing joint. The fabricated array was then demodulated using a grating interrogator (LGI-100B, Sentek Instrument, Blacksburg, VA, USA), using the wavelength scanning time division multiplexing (WSTDM) technique. This technique can accurately measure the central wavelength and reflection spectrum of each grating. Figure 10a,b show the spectra of 10 selected gratings located from 0 m to 100 m, with an interval of 10 m. Figure 10c plots the center wavelengths of each reflection peak in the FBG array, and the fluctuation was limited to less than 0.1 nm. Figure 10d shows the intensity of each reflection peak in the FBG array. The reflection peak intensity of mode LP_01_ was as high as around −40 dB, with a fluctuation of less than 5 dB: A similar level was found in a single-mode FBG array [22]. The intensity of the reflection peak of the LP_11_ mode was prone to external influences, and therefore, the fluctuation ramps up to ~8 dB. As seen in Figure 10d, the fluctuation of the coupled mode was unstable, with a value of ~16 dB. It is clear that the measurements with both the LP_01_ and LP_11_ modes have good consistency and stability, making the TM-FBG array an effective sensing network for real applications.

The 100-meter-long TM-FBG array was further tested for quasi-distributed temperature sensing. The FBG array was placed into a chamber with temperature control, and the center wavelength of the LP_01_ reflection was recorded. Figure 11a shows the variation of central wavelength when all FBGs were heated from 30 °C to 100 °C with a step of 10 °C. It can be seen that there was a slight red-shift. Figure 11b extracts the wavelength variation of the FBGs every 10 m, and we found that the wavelength shift and the change of temperature kept linear, as ~10.9 pm/°C.

As discussed in Section 4, the reflected peak intensity of the LP_11_ mode has a linear relationship with the curvature when it is larger than 25 m^−1^ (corresponding to the bending diameter 80 mm). We then designed an experiment to verify the system for quasi-distributed curvature sensing. The setup is shown in Figure 12a. The FBG array was helically winded over a stick with a diameter of 50 mm. The stick used for curvature sensing was made from polyethylene and propylene (PEP), which is the combination of polyethylene glycol (PEG) and propylene oxide (PO). The adjustable fixture clamped the winded part, applying pressure in this region. The influence of the fixture pressure is depicted in Figure 12b. When the pressure was applied, the sponge substrate was pressed and the curvature of the fiber increased rapidly towards the short axis, leading to a high bending loss. The vertex of the short axis had the maximum radius of curvature, with the bending radius defined as *R* = *a*^2^/*b* (*a* is the length of the long axis and *b* is the length of the short axis). The spatial resolution of this system can be adjusted by changing the winding density of fiber over the sponge substrate, and the sensitive threshold can be controlled by changing the diameter of the substrate material.

Figure 13a shows the results of different bending degrees at a certain position. This position was chosen between the 4th and 5th FBG (corresponding to the area from 4 m to 5 m of the TM-FBG array). The bending degree of the fiber increased by reducing the length of the short axis. When the bending curvature increased, the intensity of the reflection peaks, located at 1 m, 2 m, 3 m and 4 m, were almost unchanged. However, from 5 m to the rest of the grating array, the measured reflection intensity significantly decreased with the increase of the deformation. It was therefore determined that the position of the deformation was located between 4 m and 5 m of the fiber. Figure 13b presents the reflection intensity of the FBGs at 4 m and 5 m when the deformation increased. It can be seen that the FBG at 4 m was hardly influenced, while the peak intensity of the FBG at 5 m was reduced. When the maximum curvature changed from 50 m^−1^ to 300 m^−1^, the reflected peak intensity of the 5th FBG decreased at a rate of −0.056 dB/m^−1^, and the linearity was 0.991. Note that in the measurements, the first few data points (marked with red circles in Figure 12b) failed to meet the expected trend. It was supposed that when the bending degree was low, errors could be large due to the tension raised during fiber winding.

As shown in Figure 14a, six positions were selected randomly and individual deformation was then applied. The measured results show that the first FBGs of the reflection peak intensity decreased were located at 5 m, 6 m, 10 m, 12 m, 16 m and 19 m. In general, this system can reliably identify the positions of the deformation spots. Moreover, to test the multi-position measurements, five positions were simultaneously applied the deformation at 10 mm, and the result is shown in Figure 14b. The deformation regions can be identified at 6 m to 7 m, 8 m to 9 m, 13 m to 14 m, 16 m to 17 m and 19 m to 20 m.

The sensitivity of the temperature sensing was measured ~10.9 pm/°C within the range from 30 °C to 100 °C. For the curvature sensing, the sensitivity reached −0.056 dB/m^−1^ in the range 50 m^−^^1^ to 300 m^−^^1^. We used the central wavelength of the reflection peak of LP_01_ mode and LP_11_ mode to realize the temperature and measure curvature. Since the two physical modes were completely independent, therefore, dual-parameter measurement could be realized at the same time. Compared with an ordinary grating array, the current system could realize real-time temperature sensing, as well as a fast response to curvature changing. Such a system is suitable for being used as an alarm system to measure and locate one region with larger curvature, and monitor the temperature continuously.

## 6. Conclusions

In this paper, mode characteristics of a fiber with different diameters from 125 μm to 240 μm were theoretically investigated for the purpose of figuring out the conditions for two-mode fiber. The proposed fiber had a diameter of 200 μm and only two transverse modes (LP_01_ and LP_11_) were supported. Using the on-line grating fabrication system, a TM-FBG array was successfully written and this TM-FBG array was further investigated for temperature and curvature sensing. The response to temperature was measured 11.2 pm/°C and the curvature response was −0.21 dB/m^−1^. Further, the TM-FBG array was analyzed using a grating interrogator. For quasi-distributed temperature sensing, the grating array performed similarly as a standard grating array, however, it possessed extensions for curvature measurements. The TM-FBG array could reliably determine the position and degree of bending along a structured object. It is believed, therefore, that such a system has potentials in many aspects regarding safety production or structural surveillance.

## Figures and Tables

**Figure 1 materials-12-01263-f001:**
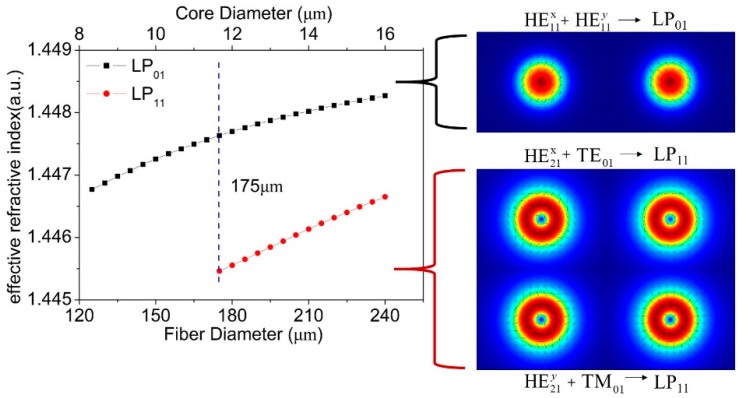
The relation between the transmission mode and diameter of the fiber the insets are mode fields of vector modes in two-mode fiber and the relationship of vector modes and LP modes.

**Figure 2 materials-12-01263-f002:**
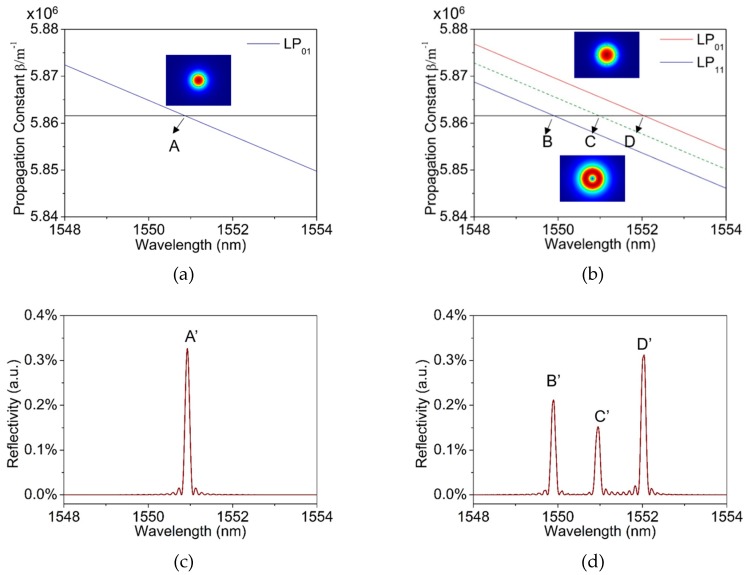
The formation mechanism of fiber Bragg grating (FBG) in a single-mode fiber and a two-mode fiber, the horizontal line (in black) refers to the propagation constant and the colorful curves depict the resonance condition. (**a**) The intersection refers A to FBG resonance condition in single-mode fiber, the inset plots the mode field of LP_01_ mode; (**b**) the intersections, B, C and D, refer to resonance conditions for LP_01_ mode, LP_11_ mode and the coupled mode in two-mode fiber, the insets plot the mode fields of LP_01_ and LP_11_ modes; (**c**) reflection spectrum of the single-mode FBG; and (**d**) reflection spectrum of the two-mode FBG.

**Figure 3 materials-12-01263-f003:**
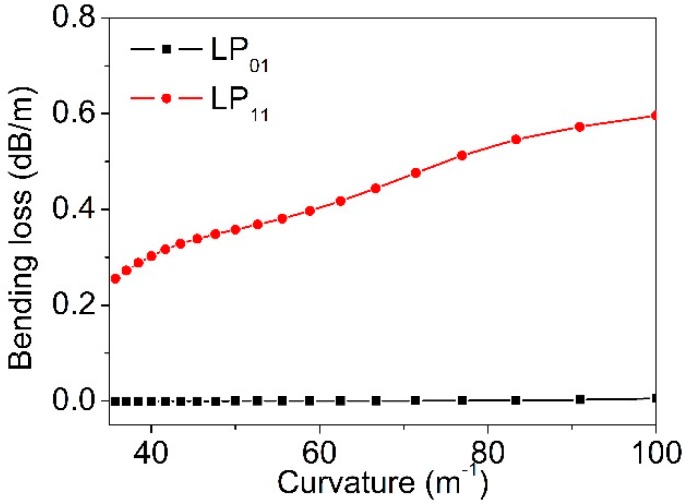
The simulation of two modes: Confinement loss when the curvature increases.

**Figure 4 materials-12-01263-f004:**
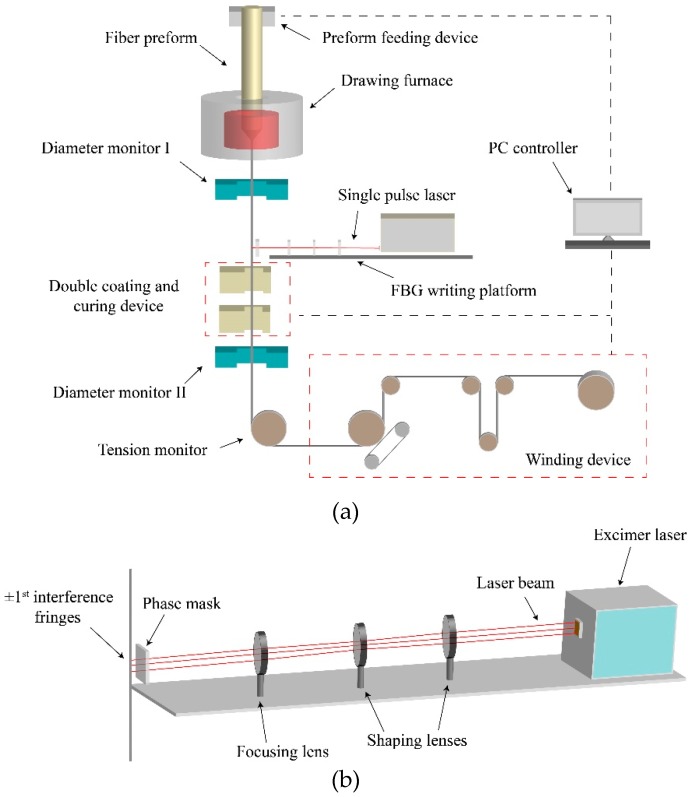
(**a**) Schematic diagram of the FBG online fabrication processes on our customized drawing tower. (**b**) Schematic diagram of the phase mask writing platform.

**Figure 5 materials-12-01263-f005:**
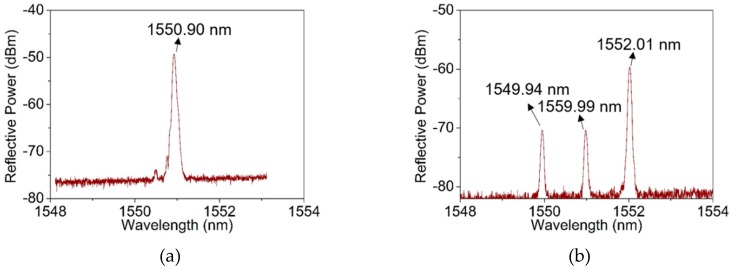
The experimental reflection spectrum of the FBG with the diameter of the fiber is (**a**) 125 μm and (**b**) 200 μm, the inset is the experimental optical field distribution.

**Figure 6 materials-12-01263-f006:**
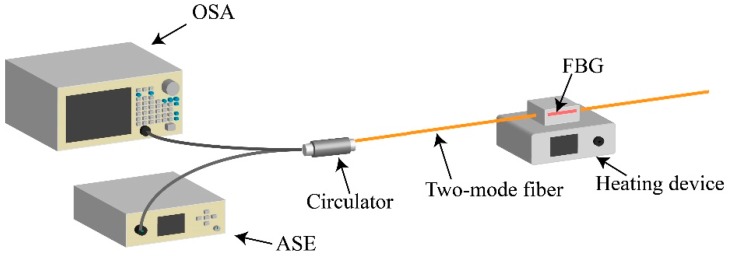
Schematic diagram of the two-mode FBG temperature experiment.

**Figure 7 materials-12-01263-f007:**
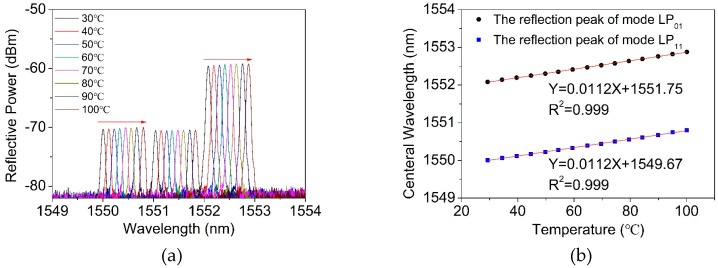
The experimental (**a**) reflection spectra and (**b**) wavelength shift of two-mode FBG with the temperature change.

**Figure 8 materials-12-01263-f008:**
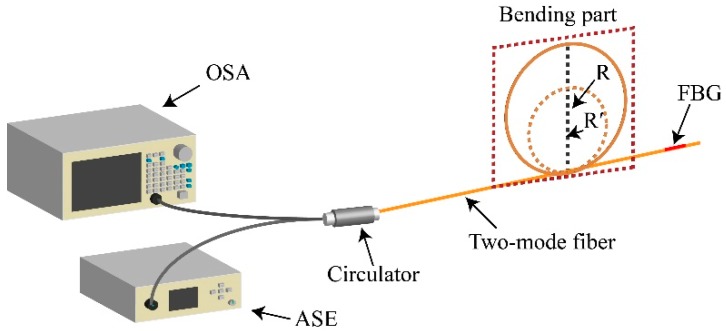
Schematic diagram of the two-mode FBG bending experiment.

**Figure 9 materials-12-01263-f009:**
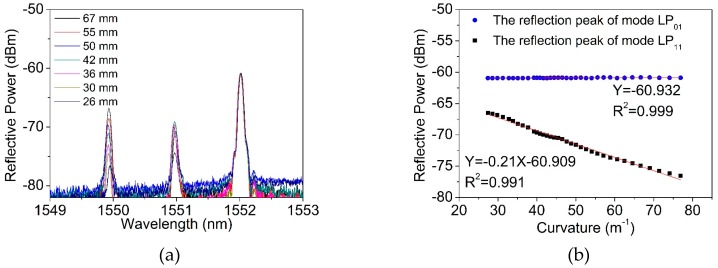
The experimental (**a**) reflection spectra, (**b**) peak intensity, (**c**) the difference of reflectivity intensity of two-mode FBG and (**d**) the transmission mode field with the curvature change.

**Figure 10 materials-12-01263-f010:**
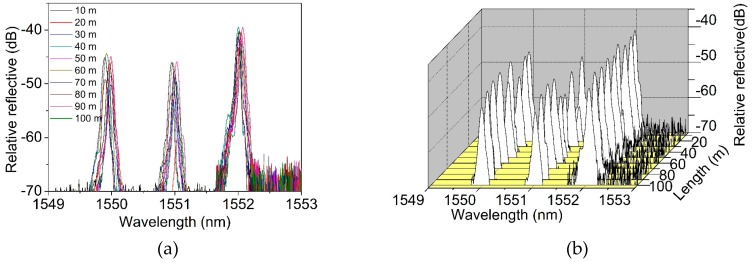
The measurements of two-mode FBG array for (**a**) and (**b**) spectra, (**c**) center wavelength, (**d**) reflection peak intensity.

**Figure 11 materials-12-01263-f011:**
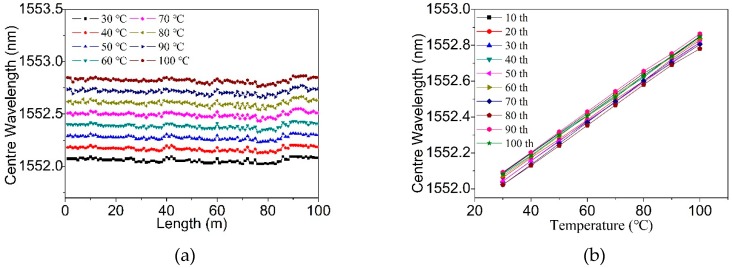
Variation of the center wavelength of (**a**) all FBGs and (**b**) selected FBGs when the temperature increase from 30 °C to 100 °C.

**Figure 12 materials-12-01263-f012:**
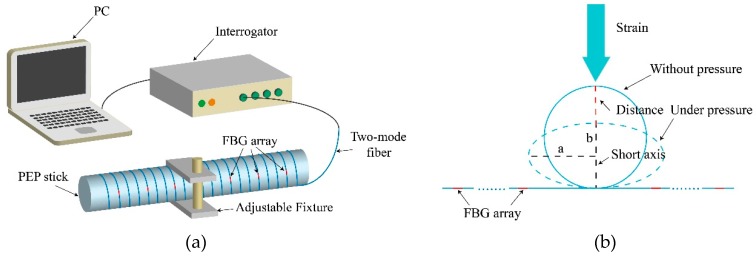
Schematic diagram of (**a**) the FBG array in the application of large curvature sensing; (**b**) the fiber is deformed by pressure.

**Figure 13 materials-12-01263-f013:**
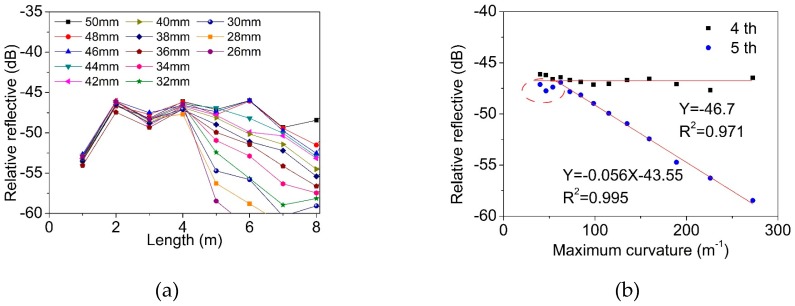
Relative change of reflection peak intensity for (**a**) all FBG and (**b**) single FBG when the axial strain distance from 0 mm to 24 mm.

**Figure 14 materials-12-01263-f014:**
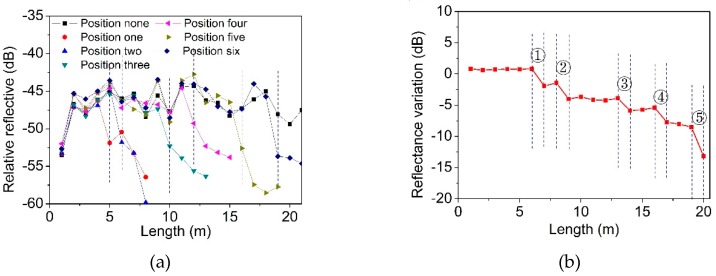
Variation of reflection peak intensity when the deformation is located in (**a**) a different area and (**b**) multiple areas.

**Table 1 materials-12-01263-t001:** Several lowest order linear polarization (LP) modes and their corresponding cut-off frequency *V_c_*.

Cut-off *V_c_*	LP mode
<2.405	only LP_01_
2.405	LP_11_
3.823	LP_02_, LP_21_
5.136	LP_31_

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
