# Peer review of "On-line Writing of Fiber Bragg Grating Array on a Two-mode Optical Fiber for Sensing Applications"

_materials, 2019, doi:10.3390/ma12081263_

Reviewer 1 Report

The authors  propose Bragg grating array on a  two-mode optical fiber for sensing applications. Present a theoretical and experimental study. The FBG is a very kown subjet with several applications in sensing. In the manuscript, the authors don´n show a clear advantage of system of writing FBG “on line”. The manuscript don’t response to:

·  Advantage of this system. Why not used comercial fiber

·  Concentration of germanium in the preform?

·  Why use a fiber core diameter of 200 um?

·  The distance between FBG.

·  Applications of this fiber. How integrate this fiber and standard fiber.

In present form i recommend reject the manuscript

Author Response

Response to Reviewer: On-line writing of fiber Bragg grating array on a two-mode optical fiber for sensing applications

Reviewer 1

Comments and Suggestions for Authors

The authors propose Bragg grating array on a two-mode optical fiber for sensing applications. Present a theoretical and experimental study. The FBG is a very known subject with several applications in sensing. In the manuscript, the authors don’t show a clear advantage of system of writing FBG “on line”. The manuscript doesn’t response to:

Point 1: Advantage of this system. Why not used commercial fiber 

Response 1: Thanks for the comments. We add the advantages of on-line writing technique in the manuscript as follows:

Using the on-line writing technique, a large number of FBGs can be continuously inscribed in a single fiber piece, without complex post-processing steps such as coating decortication or fiber-splicing, as required by conventional FBG writing. This technique brings advantages such as highly consistent grating reflectivity, increased mechanical strength and reduced loss.

Different from the commercial fiber with only one transmission mode, LP01 mode, the two mode fiber we used can realized the co-existence of the LP01 and LP11 modes with low loss. Using the different sensing response of the LP01 and LP11 modes, the TM-FBGs can realize the quasi-distributed sensing of temperature and curvature simultaneously.

Point 2: Concentration of germanium in the preform?

Response 2: Thanks, we add the following details in the text:

The optical fiber preform for FBG inscription is the glass preform, with low amount of germanium, less than 0.6%, doped in the fiber core, and the numerical aperture (NA) of 0.14.

Point 3: Why use a fiber core diameter of 200 um?

Response 3: The reason is given below:

The preform used for fiber drawing is a standard one for SMF with a fiber core diameter 8.3 um, and we used the same preform for the fabrication of the two-mode (TM) fiber. Based on our theoretical calculations, it was found for a fiber to support two modes, the core diameter needs to be expanded to 13.3 um, so as the fiber outer diameter to be ~200 um.

Point 4: The distance between FBG.

Response 4: The grating spacing of the FBG array is 1 m.

Point 5: Applications of this fiber. How integrate this fiber and standard fiber.

Response 5:

In this study, this fiber can realize the quasi-distributed sensing of temperature and curvature simultaneously. By using the fiber splicer, the two mode fiber was spliced to SMF with accurate centre alignment and low loss. After effective splicing, we can further measure the properties of TM-FBG by optical spectrum analyzer or demodulator.

Reviewer 2 Report

This paper reports about the on-line development of a Fiber Bragg Grating (FBG) array in a two-mode (TM) fiber. The behavior of TM-FBG is studied and tested for the measurement of temperature and curvature, based on the different response of LP01 and LP11 modes to such parameters. Finally, the curvature sensing by using a 100 m long fiber with one hundred FBGs is performed.

The experimentation conducted by the authors is interesting and the conclusions are supported by the results. Anyway, I have some comments/questions before the paper can be accepted for publication:

1.       The manuscript would benefit from a review of the English language.

2.       Which is the novelty of the proposed fabrication technique? Since it is not the first time that TM-FBG are fabricated.

3.       Some sentences are unclear and should be double checked/rephrased: lines 30-31, lines 31-32, lines 80-82.

4.       Add legend in Fig. 2(a) and (b).

5.       The spectra reported in Fig. 2(c) and (d) seem experimental. Please add the numerical ones or modify the text accordingly.

6.       How is the curvature simulated to obtain the results reported in Fig. 3? Please comment.

7.       Please add more details about the fibers in Section 3, for example the core diameter and NA.

8.       Line 167, I think it is Figure 9(b).

9.       The material of the stick used for curvature sensing is not clear, is it PEP (as reported on line 213) or PES (as in Fig. 12(a))? Please clarify and give full name also.

10.   Check “a” and “b” definition on lines 218-219.

Author Response

Response to Reviewer: On-line writing of fiber Bragg grating array on a two-mode optical fiber for sensing applications

Reviewer 2

Comments and Suggestions for Authors

This paper reports about the on-line development of a Fiber Bragg Grating (FBG) array in a two-mode (TM) fiber. The behavior of TM-FBG is studied and tested for the measurement of temperature and curvature, based on the different response of LP01 and LP11 modes to such parameters. Finally, the curvature sensing by using a 100 m long fiber with one hundred FBGs is performed.

The experimentation conducted by the authors is interesting and the conclusions are supported by the results. Anyway, I have some comments/questions before the paper can be accepted for publication:

Point 1: The manuscript would benefit from a review of the English language.

Response 1: Thanks very much for the positive comments, we have further polished the language of our manuscript.

Point 2: Which is the novelty of the proposed fabrication technique? Since it is not the first time that TM-FBG are fabricated.

Response 2: We add the advantages of on-line writing technique in the manuscript as follows:

Using the on-line writing technique, a large number of FBGs can be continuously inscribed in a single fiber piece of a long distance (e.g. 50 km), without complex post-processing steps such as coating decortication, grating inscription and fiber-splicing, as required by conventional FBG writing. This technique brings advantages such as highly consistent grating reflectivity, increased mechanical strength, and reduced splicing loss.

Point 3: Some sentences are unclear and should be double checked/rephrased: lines 30-31, lines 31-32, lines 80-82.

Response 3: Corrected:

Not only limited to SMFs, grating structures, in principle, can be written on any fiber with a sensitive core, for instance, a few-mode fiber. A few-mode fiber supports a limited number of transverse modes [7].

According to the calculations (Figure 2a), the propagation constant of the LP01 mode decreases monotonically with the increase of wavelength, while the grating resonance is independent of the wavelength change.

Point 4: Add legend in Fig. 2(a) and (b).

Response 4: Thanks for your advice, we replotted Figs. 2(a) and (b) as follows:

(a)

(b)

Figure 2. The formation mechanism of FBG in a single-mode fiber and a two-mode fiber, the horizontal line (in black) refers to the propagation constant, and the colorful curves depict the resonance condition. (a) the intersection refers A to FBG resonance condition in single-mode fiber, the inset plots the mode field of LP01 mode; (b) the intersections, B, C, and D, refer to resonance conditions for LP01 mode, LP11 mode and the coupled mode in two-mode fiber, the insets plot the mode fields of LP01 and LP11 modes. 

Point 5: The spectra reported in Fig. 2(c) and (d) seem experimental. Please add the numerical ones or modify the text accordingly.

Response 5: Thanks for your valuable comments, to put it clearly, we re-organized Fig.2 and Fig. 5, the experimental figures are shown in Fig.5, the relevant description was added in the text.

“According to the calculations (Figure 2a), the propagation constant of the LP01 mode decreases monotonically with the increase of wavelength, while the grating resonance is independent of the wavelength change. The propagation constant and the grating resonance have only one intersection (marked as A) at the wavelength of 1550.90 nm, corresponding to the central wavelength (marked as A’) of FBG in the refection spectrum, as shown in Figure 2(c). Figure 2(b) shows the relationship of calculated propagation constants and resonance condition of the LP01 mode, LP11 mode and the coupled mode in two-mode fiber, of which the grating resonance wavelength are marked as B, C, and D, corresponding to the three wavelength of TM-FBG in Fig. 2(d). Due to the co-existence of the LP01 and LP11 modes in the fiber, each mode has a specific wavelength satisfying the grating resonance condition, which are at wavelengths 1549.90 nm (B’) and 1552.00 nm (D’), respectively. In addition, mode coupling exists between the two LP modes, which forms a coupled-mode. The propagation constant of the coupled mode is approximately one half of the sum, of those the two LP modes. The resonance wavelength of the coupled mode is at around 1550.95 nm(C’).

(c)

(d)

Figure 2. The formation mechanism of FBG in a single-mode fiber and a two-mode fiber. (c) reflection spectrum of the single-mode FBG; (d) reflection spectrum of the two-mode FBG.

(a)

(b)

Figure 5. (a) The experimental reflection spectra of the FBG with the diameter of the fiber is (a) 125 μm and (b) 200 μm.

Point 6: How is the curvature simulated to obtain the results reported in Fig. 3? Please comment.

Response 6: Using the Finite Element Method (FEM) calculation in COSMOL, we analyse the mode change of two-mode optical fibers under different curvatures. The real and imaginary parts of the effective refractive index of LP01 and LP11 mode are calculated. Combing with equation (2), the calculation loss is used to reflect the bending loss of LP01 and LP11 modes.

Point 7: Please add more details about the fibers in Section 3, for example the core diameter and NA.

Response 7: We have added details in the manuscript:

The optical fiber preform for FBG inscription is the glass preform, with low amount of germanium, less than 0.6%, doped in the fiber core, and the numerical aperture (NA) of 0.14.

The preform used for fiber drawing is a standard one for SMF with a fiber core diameter 8.3 um, and we used the same preform for the fabrication of the two-mode (TM) fiber. Based on our theoretical calculations, it was found for a fiber to support two modes, the core diameter needs to be expanded to 13.3 um, so as the fiber outer diameter to be ~200 um.

Point 8: Line 167, I think it is Figure 9(b).

Response 8: Thanks for pointing this out, we have revised it.

Point 9: The material of the stick used for curvature sensing is not clear, is it PEP (as reported on line 213) or PES (as in Fig. 12(a))? Please clarify and give full name also.

Response 9: Thanks. The material of the stick is PEP. We have revised it as follows:

The stick used for curvature sensing is made from PEP, which is the combination of polyethylene glycol (PEG) and propylene oxide (PO).

Point 10: Check “a” and “b” definition on lines 218-219.

Response 10: Thanks a lot. The wrong definition has been revised.

a is the length of the long axis and b is the length of the short axis.

Reviewer 3 Report

The authors report on a sensor using FBGs inscribed in a few mode fiber. The results are interesting and the results are adequately presented. However, I understand that the authors made a wrong decision when choosing the journal. Their research topic is out of the scope of “Materials”. Therefore, recommend the rejection of the manuscript and encourage the authors to resubmit it to a different journal such as “Sensors”. In the following, I provide some suggestions that may help the authors to improve their manuscript quality.

The authors explanation of why there is a third peak in the FBG spectrum in Fig. 2d is very superficial. They say that there is a formation of a “coupled-mode”. Is this term usual? If yes, please add a reference. Otherwise, I would say that this term is very imprecise since all physics behind FBGs relies on mode couplings. Therefore, I believe that it is important that the authors adequately explains why they have a third peak in the FBG spectrum (around 1551 nm).

According to the text, I understand that the near field profiles in Fig. 5b were measured in transmission by altering the coupling conditions. If it is true, I suggest removing the near field profiles from the plotting area of Fig. 5b since it can induce the reader to think they are the near field profiles of the modes that were coupled by the FBG. I suggest including these near field profiles somewhere else.

Please estimate the resolution of the system.

Please include a discussion on the discrimination procedure of simultaneous temperature and curvature variations.

Author Response

Response to Reviewer: On-line writing of fiber Bragg grating array on a two-mode optical fiber for sensing applications

Reviewer 3

Comments and Suggestions for Authors

The authors report on a sensor using FBGs inscribed in a few mode fiber. The results are interesting and the results are adequately presented. However, I understand that the authors made a wrong decision when choosing the journal. Their research topic is out of the scope of “Materials”. Therefore, recommend the rejection of the manuscript and encourage the authors to resubmit it to a different journal such as “Sensors”. In the following, I provide some suggestions that may help the authors to improve their manuscript quality.

Response: Thank very much for the positive comments. We still believe that our work fits nicely to the journal since the current issue is specially dedicated to "Novel Optical Fibers, Devices and Applications" in Materials.

Point 1: The authors explanation of why there is a third peak in the FBG spectrum in Fig. 2d is very superficial. They say that there is a formation of a “coupled-mode”. Is this term usual? If yes, please add a reference. Otherwise, I would say that this term is very imprecise since all physics behind FBGs relies on mode couplings. Therefore, I believe that it is important that the authors adequately explain why they have a third peak in the FBG spectrum (around 1551 nm).

Response 1: Thanks for your valuable comments. We agree with you that the reason for the formation of FBG reflection peaks is based on the mode coupling theory. We make the following revision for more precise explanation:

In addition, mode coupling exists between the two LP modes, which forms a coupled-mode [19]. This coupled-mode in TM-FBG is generated by the coupling between the forward mode of LP01 mode and the backward mode of LP11 mode or the forward mode of LP11 mode and the backward mode of LP01 mode.

We add a reference:

[19] Lu, S. Xu, O. Feng, S. Analysis of radiation-mode coupling in reflective and transmissive tilted fiber Bragg gratings. J. Opt. Soc. Am., 2009, 26, 91-98.

Point 2: According to the text, I understand that the near field profiles in Fig. 5b were measured in transmission by altering the coupling conditions. If it is true, I suggest removing the near field profiles from the plotting area of Fig. 5b since it can induce the reader to think they are the near field profiles of the modes that were coupled by the FBG. I suggest including these near field profiles somewhere else.

Response 2: Apologize for the unclear statement. The mode profiles in Fig. 5b were measured in transmission by altering the coupling conditions, we replotted Fig. 5b and delete the confusion text.

(a)

(b)

Figure 5. (a) The experimental reflection spectrum of the FBG with the diameter of the fiber is (a) 125 μm and (b) 200 μm.

Point 3: Please estimate the resolution of the system.

Response 3: Thank you for your suggestion.

The sensitivity of the temperature sensing is measured ~10.9 pm/°C within the range from 30 to 100 °C. For the curvature sensing, the sensitivity reaches -0.056 dB/m-1 in the range 50 to 300 m-1.

Point 4: Please include a discussion on the discrimination procedure of simultaneous temperature and curvature variations.

Response 4: Thank you for your advice. In order to make it clear, we add in the text:

We use the central wavelength of the reflection peaks of the LP01 and LP11 mode to realize the temperature and curvature sensing. Since the two physical quantities are completely independent, therefore, dual-parameter measurement can be realized simultaneously.

Reviewer 4 Report

Your paper entitled “On-line writing of fiber Bragg grating array on a two-mode optical fiber for sensing applications” is very interesting.  The optical fiber sensing community could benefit from a fiber Bragg gratings (FBG) system that can permit real-time temperature sensing, as well as a fast response to curvature changing.  Your approach of employing on-line writing of FBG array on a 2 two-mode optical fiber for sensing applications has advantages over ordinary grating array systems.  The dual-parameter fiber sensing system allows quasi-distributed, simultaneous detection of temperature and curvature, which makes it suitable for structural health monitoring or perimeter security.

My comments/recommendations follow:

Since the fiber Bragg gratings are created during the fiber draw process, it is recommended that the fiber drawing speed and average fiber tension be specified.

Recommend that more detail be shown on the operation of the bending mechanism in Fig. 8 (e.g., was the bending done by forming fiber loops or winding the fiber on cylinders?).

I made a few suggestions to improve sentence structure (See the attached PDF document – Comments / Recommendations are in the yellow sticky notes).

Author Response

Response to Reviewer: On-line writing of fiber Bragg grating array on a two-mode optical fiber for sensing applications

Review 4

Comments and Suggestions for Authors

Your paper entitled “On-line writing of fiber Bragg grating array on a two-mode optical fiber for sensing applications” is very interesting. The optical fiber sensing community could benefit from a fiber Bragg gratings (FBG) system that can permit real-time temperature sensing, as well as a fast response to curvature changing. Your approach of employing on-line writing of FBG array on a 2 two-mode optical fiber for sensing applications has advantages over ordinary grating array systems. The dual-parameter fiber sensing system allows quasi-distributed, simultaneous detection of temperature and curvature, which makes it suitable for structural health monitoring or perimeter security.

Response: Thanks very much for the positive comments.

Point 1: My comments/recommendations follow:

Since the fiber Bragg gratings are created during the fiber draw process, it is recommended that the fiber drawing speed and average fiber tension be specified.

Response 1: Thank you for your suggestion. Details added:

By controlling the preform feeding and fiber drawing rates, the diameter of fiber was increased from 125 to 200 μm, for the reason to draw a TM fiber. To avoid excessive shaking and breaking in the grating written process, the drawing speed was limited to 10~15 m/min and the average tension was around 10~30 N.

Point 2 Recommend that more detail be shown on the operation of the bending mechanism in Fig. 8 (e.g., was the bending done by forming fiber loops or winding the fiber on cylinders?).

Response 2: Thanks for your suggestion. We add more details about Figure 8 as follows:

The experimental setup is depicted in Fig. 8. The optical fiber was wrapped around a loop with adjustable radius. The smaller the loop radius, the severer the bending applied to the TM fiber.

Point 3I made a few suggestions to improve sentence structure (See the attached PDF document – Comments / Recommendations are in the yellow sticky notes).

Response 3: Thank you for your valuable comments. We accepted and revised the manuscript accordingly.

Round  2

Reviewer 1 Report

The authors present a review version with changes that clarify previous questions. I recommned publish

Reviewer 2 Report

The revised manuscript is suitable for publication.

Reviewer 3 Report

I understand that the authors adequately answered the reviewers questions and explained why their manuscript is suitable for Materials. I believe, therefore, that the manuscript can now be considered for publication.